# Multi-Temporal SAR Data Large-Scale Crop Mapping Based on U-Net Model

**Sisi Wei [1,2], Hong Zhang [1,*], Chao Wang [1,2,*], Yuanyuan Wang [1,2] and Lu Xu [1,2]**

1   Key Laboratory of Digital Earth Science, Institute of Remote Sensing and Digital Earth,
    Chinese Academy of Sciences, Beijing 100094, China; weiss@radi.ac.cn (S.W.);
    wangyy2016@radi.ac.cn (Y.W.); xulu@radi.ac.cn (L.X.)
2   College of Resources and Environment, University of Chinese Academy of Sciences, Beijing 100049, China
*   Correspondence: zhanghong@radi.ac.cn (H.Z.); wangchao@radi.ac.cn (C.W.); Tel.: +86-10-8217-8186 (H.Z.)

**Abstract:** Due to the unique advantages of microwave detection, such as its low restriction from the atmosphere and its capability to obtain structural information about ground targets, synthetic aperture radar (SAR) is increasingly used in agricultural observations. However, while SAR data has shown great potential for large-scale crop mapping, there have been few studies on the use of SAR images for large-scale multispecies crop classification at present. In this paper, a large-scale crop mapping method using multi-temporal dual-polarization SAR data was proposed. To reduce multi-temporal SAR data redundancy, a multi-temporal images optimization method based on analysis of variance (ANOVA) and Jeffries–Matusita (J–M) distance was applied to the time series of images after preprocessing to select the optimal images. Facing the challenges from smallholder farming modes, which caused the complex crop planting patterns in the study area, U-Net, an improved fully convolutional network (FCN), was used to predict the different crop types. In addition, the batch normalization (BN) algorithm was introduced to the U-Net model to solve the problem of a large number of crops and unbalanced sample numbers, which had greatly improved the efficiency of network training. Finally, we conducted experiments using multi-temporal Sentinel-1 data from Fuyu City, Jilin Province, China in 2017, and we obtained crop mapping results with an overall accuracy of 85% as well as a Kappa coefficient of 0.82. Compared with the traditional machine learning methods (e.g., random forest (RF) and support vector machine (SVM)), the proposed method can still achieve better classification performance under the condition of a complex crop planting structure.

**Keywords:** crop mapping; SAR; Sentinel-1; multi-temporal; U-Net

## 1. Introduction

China supplies 21% of the world's population with only 7% of the world's arable land. However, with the development of agricultural modernization in recent years, problems in agricultural development in China, such as the agricultural foundation, are weak; the quality and safety of agricultural products are more problematic, and the structural imbalance of agricultural production as well as its agricultural benefits are relatively low, even if having become more prominent. These problems restrict the development of China's agriculture [1]. Therefore, strengthening the monitoring of the current situation of agriculture and the timely formulation of scientific and reasonable policies to improve agricultural development can effectively promote the development of agriculture in China [2].

Remote sensing technology has become one of the main means of extracting crop information because it can obtain crop growth and change information quickly and accurately [3]. Many achievements have been made in traditional crop identification and area monitoring based on optical remote sensing data, and both theory and technology have grown substantially [4]. However, in

practical applications, optical data are vulnerable to weather conditions, such as clouds and rain; thus, it is often impossible to obtain images of the critical period of crop growth, which limits the application of optical remote sensing technology in agriculture [5].

Synthetic aperture radar (SAR), as a means of remote sensing observation that is unaffected by weather and time, has shown obvious advantages in the field of surface observation [6,7]. With the development of space-borne SAR technology, a large number of SAR data are available for land observation, and the application of SAR data in agriculture is becoming more extensive [8,9]. At present, crop mapping based on SAR data mainly uses backscattering, polarization, and time series features of multi-polarization and multi-temporal SAR data for crop identification [10,11]. Common SAR data sources include ENVISAT/ASAR [12], Cosmo-SkyMed [13], TerraSAR-X [14], RADARSAT-2 [15], ALOS-2/PALSAR-2 [16], and Sentinel-1 [17,18]. To improve the accuracy of crop classification, researchers introduced support vector machine (SVM) [19], random forest (RF) [20] and other machine learning methods into SAR crop recognition, thus effectively improving the accuracy of crop classification [21]. However, the classification accuracy of the shallow structure model [22] is not satisfactory when facing a complex crop planting structure. With the advent of the big data era and the improvement of scientific computing, including cloud computing, parallel computing, and graphics processing unit (GPU) optimization, deep learning technology has developed rapidly. Deep learning attracts the attention of SAR crop classification researchers, since it shows good classification accuracy and efficiency in optical images, and can learn high-level context features through a large number of neurons, which overcomes many limitations of traditional classification methods. Hirose et al. used a complex-valued convolution neural network (CV-CNN) and a reinforcement learning model to conduct a lot of pioneering work on land use classification of SAR [23]. Using multi-temporal Landsat-8 optical data and Sentinel-1 SAR data, Kussul et al. applied a multi-level deep learning network to crop mapping with a complex crop structure in Ukraine [24]. Castro et al. used an automatic encoder (AE) and a convolutional neural network (CNN) to classify crop from multi-temporal optical data and SAR data. The experimental results show that the overall classification accuracy of CNN and AE is better than that of traditional classification methods [25]. Ndikumana et al. analyzed Sentinel-1 time-series data in Camargue, France, based on a deep recursive neural network (RNN), and found that the classification results of two RNN-based classifiers were significantly better than those of classical methods [26].

In recent years, with the capability to learning hierarchical features, the fully convolutional network (FCN) has made substantial progress in the field of image semantic segmentation [27]. Due to the similarity between semantic segmentation in computer vision and remote-sensing image feature classification, researchers began to introduce the FCN to learn the global neighborhood features of remote sensing image pixels. An FCN is an end-to-end deep supervisory network structure that expands the perceptual domain by convolutional layer downsampling, increases context information, and improves classification accuracy. In addition, by adding an upsampling layer, the sizes of the output image and the input image were the same. The dimensions are consistent and achieve pixel-by-pixel classification. Currently, FCNs are used mostly in high-resolution optical images [28] and full-polarized SAR images [29] with the main application of extracting a single class of ground objects, and it has achieved higher classification accuracy than traditional methods [30]. In 2015, based on FCNs, Ronneberger et al. proposed a convolutional network for biomedical image segmentation, U-Net [31]. Compared with an FCN, a U-Net model is more suitable for multi-channel remote sensing data (channel number > 3) classification, and it can better overcome the problem of small sample size and unbalanced sample size. In the past two years, researchers have begun to apply the U-Net model to multi-channel remote sensing data classification. Zhang et al. combined the characteristics of U-Net and residual learning methods to achieve the extraction of road information [32]. Xu et al. used the residual U-Net structure to extract the building information of urban areas, and combined the guided filtering to post-process the images to obtain better extraction results [33].

Deep learning semantic segmentation technology has great potential to solve the problem of difficulty in improving crop classification accuracy when the planting structure is complex, but research in this field has been rarely explored. Therefore, this paper attempts to apply an improved FCN model, the U-Net, to multi-temporal SAR data in order to achieve a high-precision extraction of large-area and multi-type crops.

## 2. Methods

In this paper, a multi-temporal SAR data large-scale crop mapping algorithm based on the U-Net model was proposed, and the flow of our method is shown in Figure 1. First, all the multi-temporal SAR data were preprocessed, including data import, multi-looking, coregistration, multi-temporal filtering, geocoding, and calibration. Then, a multi-temporal images optimization method based on the analysis of variance (ANOVA) and Jeffries–Matusita (J–M) distance was introduced to reduce multi-temporal data redundancy. Based on optical data and field investigation, a network training sample set was developed, and the diversity of samples was enhanced by geometric transformations (cutting, rotation, and flipping). Finally, a well-trained U-Net model using the multi-temporal SAR sample set was used to achieve the large-scale crop mapping.

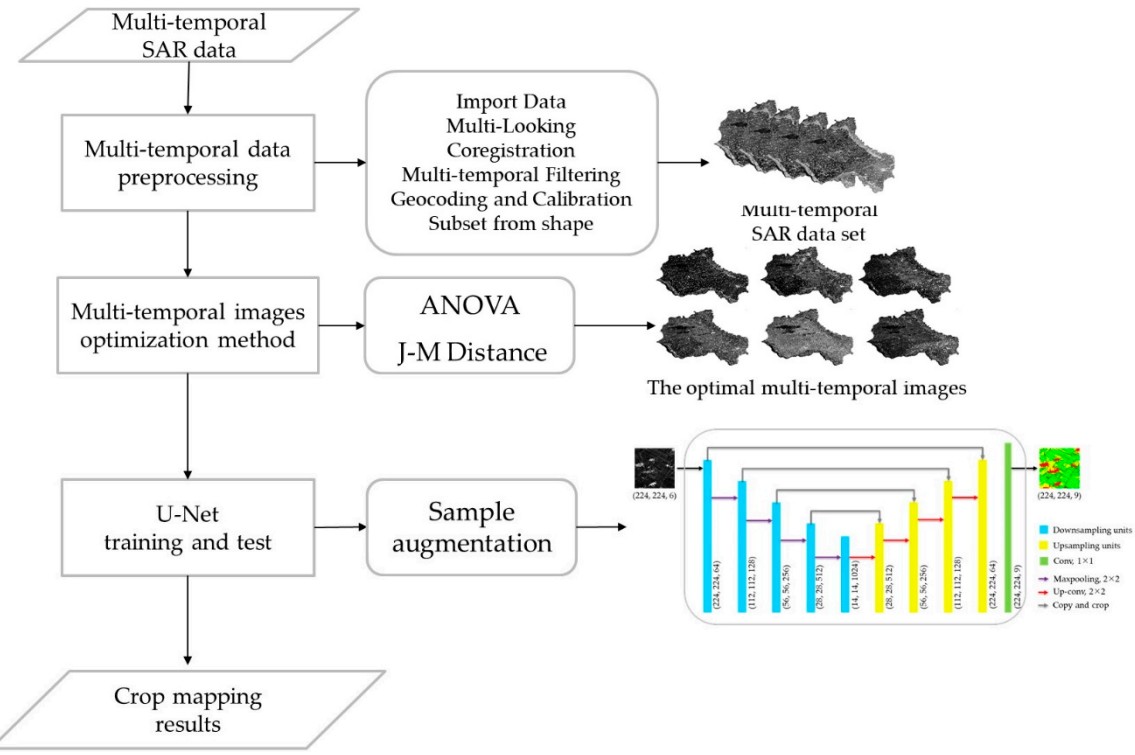

**Figure 1.** Flow chart of Multi-temporal SAR data crop mapping method.

### 2.1. Multi-Temporal Images Optimization Method

#### 2.1.1. Analysis of Variance

ANOVA, a statistical method, is used for the significance test between multiple groups of means [34]. The basic idea was to decompose the total variation among all observations into multiple parts according to the source of its variation. Comparing the variance of different source variations, it was inferred by F-distribution statistics whether a certain factor has an influence on the observation index [35]. This paper uses a one-way analysis of variance to determine the validity of each phase of the data to identify different crop types. Its calculation method is as follows.

Under a certain phase, it is assumed that there is a total of m types of samples, that each of which randomly selects n samples, and that the total number of samples is $N(N = m \times n)$. The hypothesis of *F*-test $H_0$: There is no significant difference between different samples; that is, $\mu_1 = \mu_2 = \cdots = \mu_m$. *F* is defined as Formula (1).

$$F = (\frac{S_A}{DB})/(\frac{S_E}{DW}) \tag{1}$$

$$S_A = \sum_{i=1}^{m} n_i(\mu_i - \mu)^2 \tag{2}$$

$$DB = m - 1 \tag{3}$$

$$S_E = \sum_{i=1}^{m}\sum_{j=1}^{n}(x_{ij} - \mu_i)^2 \tag{4}$$

$$DT = N - 1 \tag{5}$$

where $S_A$ is the sum of squares between groups, which is the difference between the mean value of each type of samples and the total sample data, and DB is the degree of freedom of $S_A$. $S_E$ is the sum of squares within group, which is the difference between the observation value of the backscattering coefficient of the same type of ground object, and DW is the degree of freedom of $S_E$. In addition, $x_{ij}$ is the observed value of the backscattering coefficient of the sample, $\mu$ is the mean of the overall sample, $\mu_i$ is the mean of the *i* sample set.

For a given significance level $\alpha$, the critical value $F_\alpha(DB, DW)$ is determined from the *F* distribution table. If $F > F_\alpha(DB, DW)$, $H_0$ is rejected, that is, there is a significant difference between samples from different types; otherwise, $H_0$ is accepted, meaning there is no significant difference between the samples, and the quantity of samples cannot be used for crop classification [36].

### 2.1.2. Jeffries–Matusita Distance

The J–M distance is a distinguishing indicator of the separability between samples in the remote sensing field [37]. The J–M distance ranges from 0 to 2. The larger the value, the better the separability between different sample categories [38]. Its calculation formula can be expressed as Formula (8).

$$\mathrm{JM}(C_i, C_j) = \int \left[\int [\sqrt{p(\frac{X}{C_i})} - \sqrt{p(\frac{X}{C_j})}]\right]^2 dX \tag{6}$$

where $p(\frac{X}{C_i})$ represents the probability that the i pixel belongs to the $C_i$ class.

### 2.2. U-Net

FCN is the pioneering work of deep learning in semantic segmentation, which improves the application performance of classical CNN model on pixel-level image classification [39]. The significant advantage of FCN is the end-to-end segmentation, but the disadvantage is that the segmentation results are not good enough [40]. U-Net was improved based on the FCN, and data augmentation can be used for network training with a small amount of sample data [31]. The main structure of U-Net is similar to the letter U. The whole network consists of mainly two parts, including a contracting path and an expansive path. The contracting path is used mainly to capture the context information in the image, and the expansive path is used to accurately localize the part that needs to be segmented. To accurately locate the part, the high-pixel features extracted from the contracting path are combined with the new feature map during the upsampling process to maximally preserve the most important feature information in the process of downsampling. Furthermore, to enable more efficient operation of the network structure, there are no fully connected layers in the structure, which can greatly reduce the parameters that need to be trained so that the neural network can be successfully trained with small data.

This paper uses the U-Net network architecture as shown in Figure 2, which has the same kernel function size, stride, activation function of the convolutional layer, pooling layer, and deconvolution layer as the network proposed in Ronneberger's article [31]. It includes a contracting path (blue part) and an expansive path (yellow part). Every step in the contracting path consists of two $3 \times 3$ convolutions (with padding and a rectified linear unit (ReLU)) and a $2 \times 2$ max pooling operation to downsample the input images. Furthermore, each downsampling step will double the number of feature channels. Every step in the expansive path consists of a $1 \times 1$ deconvolution (the activation function is also ReLU) and two $3 \times 3$ convolutions. The feature map from the corresponding contracting path will be added to the upsampling at each step to restore image details. The last layer of the network is a $1 \times 1$ convolution layer, through which a 64-channel feature map can be converted to the number of required classification results. In total, the network includes 23 convolutional layers.

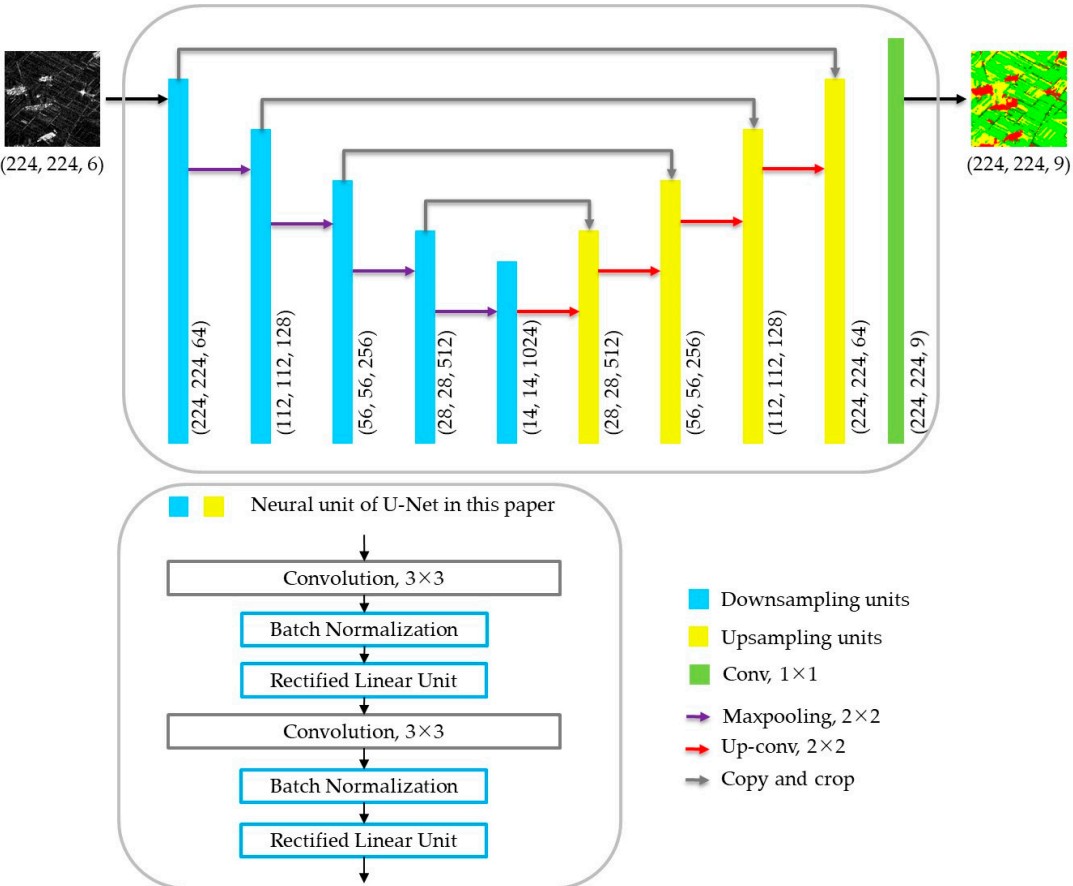

**Figure 2.** The U-Net architecture of this paper. Compared with U-Net [31], batch normalization algorithm is introduced to every convolution to improve the network training efficiency.

However, U-Net was originally proposed to solve biomedical imaging problems, which is a network for binary segmentation of biomedical images [41]. To apply U-Net to crop classification, the U-Net architecture adopted in this paper is slightly different from classic U-net in Ronneberger's article, as follows.

(1) Considering the number of various types of crop in the training sample set is uneven, which is caused by the disproportion acreage of different crops in actual production, this paper introduces the BN algorithm [42] into the U-Net network model, that is, adding a batch normalization layer between convolution layer and ReLU in each neural unit of U-Net, to improve the network training efficiency. The BN algorithm is a simple and efficient method to improve the performance of neural networks proposed by Ioffe and Szegedy in 2015 [42]. BN acts on the input of each

neural unit activation function (such as sigmoid or ReLU function) during training to ensure the input of activation function can satisfy the distribution of mean is 0 and variance is 1 based on each batch of training samples. For a value $x_i$ in a batch of data, an initial BN formula such as Formula (7).

$$BN_{initial}(x_i) = \frac{x_i - \mu_B}{\sqrt{\sigma_B^2 + \epsilon}} \tag{7}$$

In Formula (7), BN restricts the input of activation function to normal distribution, which restricts the expressive capacity of network layer. Therefore, a new proportional parameter $\gamma$ and a new displacement parameter $\beta$ are added to the BN formula. Both $\gamma$ and $\beta$ are learnable parameters. Finally, batch standardized formulas for deep learning networks are obtained, such as Formula (8).

$$\text{BN}(x_i) = \gamma \left( \frac{x_i - \mu_B}{\sqrt{\sigma_B^2 + \epsilon}} \right) + \beta \tag{8}$$

(2)  In addition, this paper uses the U-Net network with padding to ensure that the input image and the output image size are unchanged, and the padding value is 1.

## 3. Study Area and Data

### 3.1. Fuyu City

The study region is Fuyu City, located in the northwest of Jilin Province, China (Figure 3). It belongs to the temperate zone monsoon climate. In general, the spring in Fuyu City is from 1st May to 24th June, comprising 55 days; the summer is from 24th June to 13th August, comprising 50 days; autumn is from 13th August to 30th September, comprising 47 days; and winter is from 30th September to 1st May, thus comprising 213 days. Snowfall and early ice occur during late October or early November. The stable icing period is in late November, and the average ice thickness is approximately 0.95 m. The land is frozen in mid-November, and the depth of the frozen soil is 1.3–2.0 m. Thawing occurs between late March and early April.

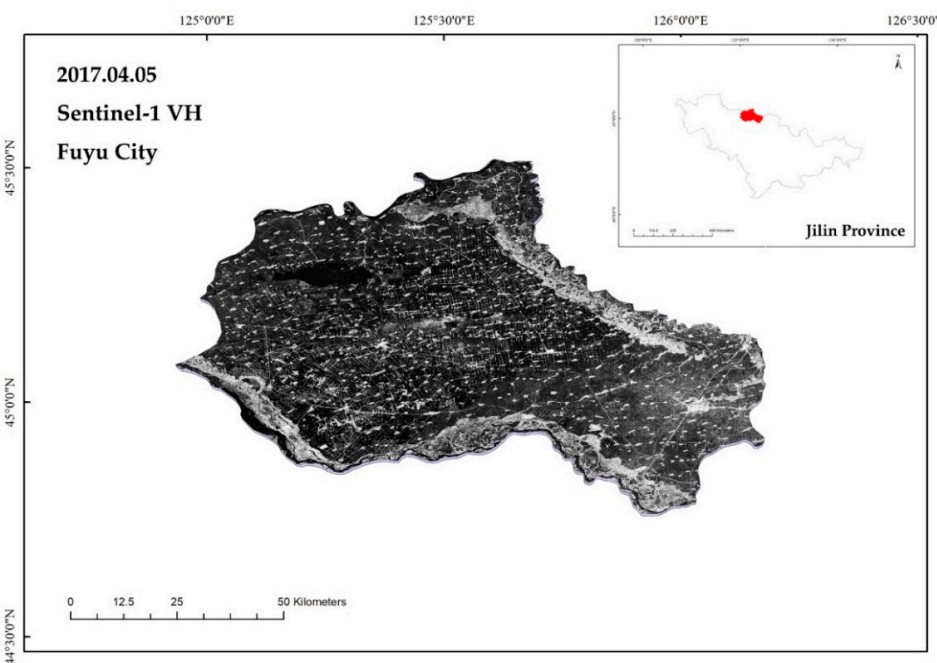

**Figure 3.** Study region: Fuyu City, Jilin Province, China.

Fuyu City has a total area of 3400.50 km$^2$ in cultivated land, including 249.42 km$^2$ of paddy fields, 9.14 km$^2$ of greenhouse land, and 3141.94 km$^2$ of dry land. In addition, we discovered that the main crops in Fuyu City include corn, peanut, soybeans, and rice, by consulting the statistical yearbook of Jilin Province (http://tjj.jl.gov.cn/). Consequently, the classification system in this paper was affirmed with four crops (including corn, peanut, soybeans, and rice,) and 4 non-crops (buildings, vegetation, water, and bare land).

*3.2. Experimental Data*

3.2.1. SAR Data

Sentinel-1 satellite can provide C-band dual-polarization SAR data with a 12-day revisit period. At the same time, its interferometric wide swath (IW) mode can provide a wide range of coverage with 250 km, which has great application prospects in large-scale crop mapping. According to the growth and development period of the main crops in Fuyu City, 18 scenes from February to October 2017 were selected in this paper to cover the Sentinel-1 IW model SLC data of Fuyu City. Table 1 shows the basic information of the experimental data.

**Table 1.** Basic information of the experimental data.

| Num. | Date | Satellite | Polarization | Orbit Direction |
|------|------|-----------|--------------|-----------------|
| 1 | 4 February 2017 | S1B | VV/VH | Descend Left-looking |
| 2 | 24 March 2017 | S1B | VV/VH | Descend Left-looking |
| 3 | 5 April 2017 | S1B | VV/VH | Descend Left-looking |
| 4 | 17 April 2017 | S1B | VV/VH | Descend Left-looking |
| 5 | 29 April 2017 | S1B | VV/VH | Descend Left-looking |
| 6 | 23 May 2017 | S1B | VV/VH | Descend Left-looking |
| 7 | 4 June 2017 | S1B | VV/VH | Descend Left-looking |
| 8 | 16 June 2017 | S1B | VV/VH | Descend Left-looking |
| 9 | 28 June 2017 | S1B | VV/VH | Descend Left-looking |
| 10 | 10 July 2017 | S1B | VV/VH | Descend Left-looking |
| 11 | 22 July 2017 | S1B | VV/VH | Descend Left-looking |
| 12 | 3 August 2017 | S1B | VV/VH | Descend Left-looking |
| 13 | 15 August 2017 | S1B | VV/VH | Descend Left-looking |
| 14 | 8 September 2017 | S1B | VV/VH | Descend Left-looking |
| 15 | 20 September 2017 | S1B | VV/VH | Descend Left-looking |
| 16 | 2 October 2017 | S1B | VV/VH | Descend Left-looking |
| 17 | 14 October 2017 | S1B | VV/VH | Descend Left-looking |
| 18 | 26 October 2017 | S1B | VV/VH | Descend Left-looking |

3.2.2. Ground Truth Data

In this paper, Landsat-8 Operational Land Imager (OLI) data in 2017 and Google Earth optical images were used as the main auxiliary data. The researchers found that corn was dark green on Landsat-8 true color images (R: Band 4, G: Band 3, B: Band 2), while soybean was bright green. Rice showed a dark red color on Landsat-8 standard false color images (R: Band 5, G: Band 4, B: Band 3). Peanuts show different characteristics from the above three crops, which shows similar colors to sandy land in true color images [43]. Table 2 shows the basic information of the Landsat-8 OLI data.

**Table 2.** Basic information of the auxiliary data.

| Num. | Date | Satellite | Sensor | Orbit Number |
|------|------|-----------|--------|--------------|
| 1 | 23 July 2017 | Landsat-8 | OLI | 118/29 |
| 2 | 24 August 2017 | Landsat-8 | OLI | 118/29 |
| 3 | 9 September 2017 | Landsat-8 | OLI | 118/29 |

In addition, the information on crop planting structures in Fuyu City was collected through field investigation. It was found that the farmland in Caijiagou Town, Sancha Town, Xinyuan Town, and Tao Laizhao Town (in the eastern part of Fuyu City) is mostly fertile chernozem, which is suitable for the large-scale planting of corn. Farmland in the central region is mostly aeolian sandy soil, which is suitable for planting peanuts. In addition, China's largest peanut trading market is located in Sanjingzi Town in the middle of Fuyu City, so Sanjingzi Town, Xinglong Town, Zengsheng Town, and Xinzhan Town in the middle of Fuyu City all cultivate peanuts. In the southeastern and northwestern areas, Tao Laizhao Town, Wujiazhan Town, Desheng Town, and Changchunling Town are all along the river and are thus more suitable for planting rice. Furthermore, we also know that the management of cultivated land in Fuyu City includes both state-owned farms and smallholders. The former generally tends to plant a single type of crop in large areas, while the latter tends to plant a variety of crops in certain areas.

Therefore, on the basis of auxiliary data and field investigation, the ground truth data of the study area was obtained through visual interpretation. Considering the spatial autocorrelation in neighboring units [44], we uniformly selected reference plots throughout the study area, and most of them were far apart, so that these plots could be considered to be mutually independent. Specially, we selected two regions with complex crop planting structures to compare the performance of different methods, and these two regions were not involved in accuracy assessment, using a confusion matrix. The ground truth image is shown in Figure 4, and the distribution of each class is shown in Table 3.

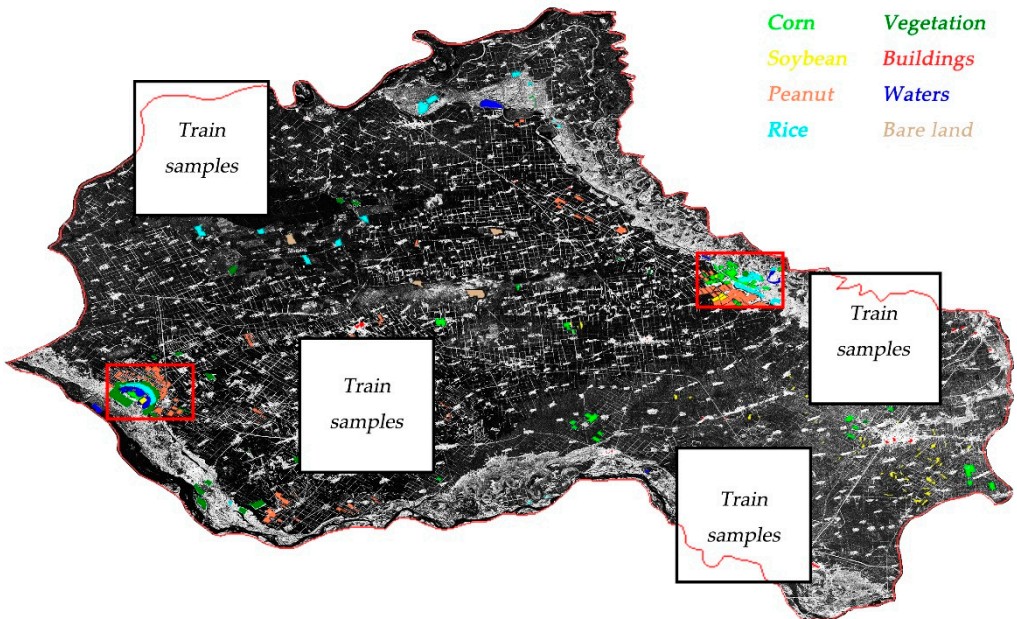

**Figure 4.** The ground truth of study area. Colored polygons represent the reference plots location of the eight types of ground objects. Two areas with complex crop planting structure are in the red rectangles. The areas in black squares are training samples for network, and there is no intersection between the training samples and the reference plots.

**Table 3.** The distribution of per class of the ground truth.

| Class | Plot Count | Pixel Count | Acreage/km$^2$ | Percent (Area)/% |
|---|---|---|---|---|
| corn | 185 | 37,607 | 9.13 | 16.99 |
| Peanut | 153 | 39,181 | 9.51 | 17.70 |
| Soybeans | 118 | 13,444 | 3.26 | 6.07 |
| Rice | 124 | 38,890 | 9.44 | 17.56 |
| Buildings | 30 | 7818 | 1.90 | 3.53 |
| Vegetation | 88 | 45,477 | 11.04 | 20.54 |
| Waters | 20 | 10,872 | 2.64 | 4.91 |
| Bare land | 30 | 28,120 | 6.83 | 12.70 |
| Total | 748 | 221409 | 53.76 | 100 |

## 4. Experimental Results

### 4.1. Results of Multi-Temporal Images Optimization

After pre-processing, two multi-temporal SAR data sets (VV/VH) were constructed with the 18 scenes of Sentinel-1 images according to the acquisition time sequence. We randomly selected 100 pixels from ground truth data for each type of ground objects, and we plotted the corresponding time-varying curves of the backscattering coefficients in dB values based on the statistical information of the samples, as shown in Figure 5.

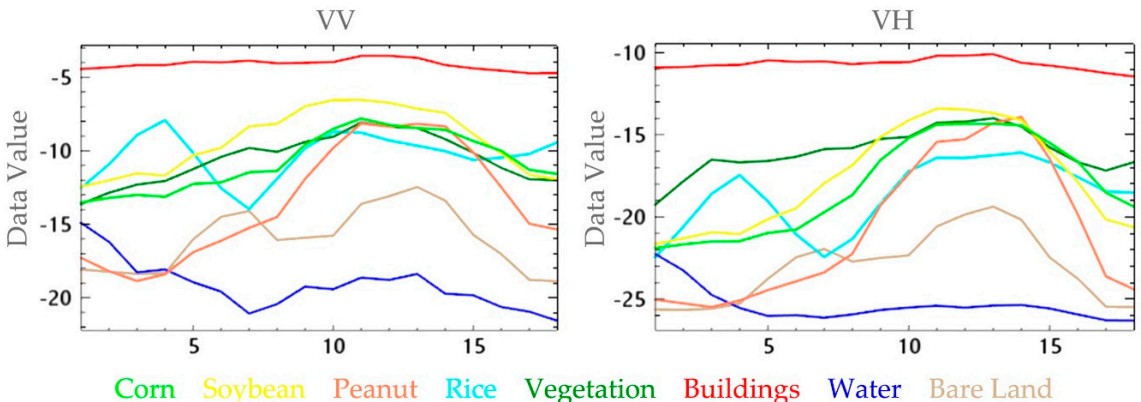

**Figure 5.** Time series curve analysis of different ground objects.

As can be seen from Figure 4: (1) the backscattering coefficient in dB values of buildings is always maintained at a high level; (2) after the thawing period of water body, the backscattering coefficient in dB values decreases obviously and remains low; (3) the backscattering coefficients in dB values of the four crops, other vegetation and bare land are time-varying, but the trends are different. It confirms the feasibility of multi-temporal SAR data for crop mapping.

However, Figure 5 shows that not every temporal image can effectively distinguish the eight types of ground objects, that is, there is redundant information in these 36 scenes of images. Within a certain range, the separability among ground object types becomes stronger with the increase of the number of temporal images. However, when the number of images is too large, the influence on the ground objects separability is small, but it will increase data redundancy and reduce the efficiency of the classification network. To this end, this paper proposes a temporal images optimization method by combining the ANOVA and the J–M distance. From each ground object contained in the study area, 10 samples for ANOVA were randomly selected. Table 4 shows the ANOVA results of the image 1 of the time-series data set.

**Table 4.** The results of ANOVA of the image 1.

| Type | SS [1] | Df [2] | MS [3] | F | α | $F_\alpha$ |
|---|---|---|---|---|---|---|
| Between group | 0.120312 | 7 | 0.017187 | | | |
| Within group | 0.023171 | 72 | 0.000322 | 53.40586 | 0.05 | 2.140 |
| Total | 0.143483 | 79 | —— | | | |

[1] SS: Sum of Square; [2] Df: Degree of freedom; [3] MS: Mean of Square.

In Table 4, The *F* value of the image 1 is 53.40586, which is much larger than the threshold $F_\alpha$ (α = 0.05) when the degree of freedom was (7, 72). After calculating the *F* detection values of all multi-temporal SAR images, it was found that all *F* values are greater than $F_\alpha$, which indicates that each image in a multi-temporal image sequence has the potential to distinguish the ground object samples. However, the *F* values of different temporal images are not equal, and the size of the *F* value reflects the ability of different temporal images to distinguish the sample of the ground objects. Therefore, the multi-temporal image is reordered according to their *F*-test values. Then the J–M distance among samples is used to characterize the relationship between the comparability of samples with the number of temporal images. It is generally believed that when the J–M distance between training samples of different land types is in the range of 1.8 to 2.0, the land class has strong separability [44].

By adding temporal images in descending order of *F* value, a new multi-temporal image was constructed and the J–M distance between samples was calculated when a temporal image was added. In addition, we also calculated the J–M distance between samples when temporal images were added in chronological order. In both cases, the relationship between the number of temporal images and the J–M distance between samples is shown in Figure 6.

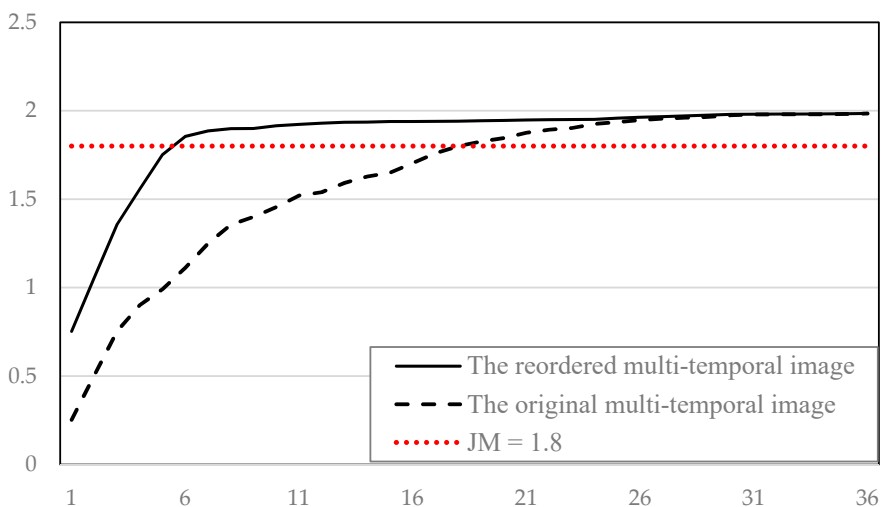

**Figure 6.** Relationship between the number of temporal images and the J–M distance.

It can be seen from Figure 6 that the J–M distance between samples increases faster when adding sequential images according to the reordered multi-temporal images compared with the original multi-temporal images. In other words, increasing the dimension of multi-temporal images in descending order of *F* value, can obtain better sample separability (J–M > 1.8) with fewer temporal images compared with in chronological order. Furthermore, we can see that the minimum value of the J–M distance increases from 0.7529 to 1.8425 when the number of temporal images increases from 1 to 6, and the minimum value of the J–M distance increased from 1.8560 to 1.9844 when the number of temporal images increases from 7 to 36, where it tends to be stable. Therefore, the temporal images which are the top six of the *F*-test values in the significant difference analysis are synthesized into a 6-channel image for the subsequent training of the U-Net network and crop classification in Fuyu City. The six selected temporal images are shown in Figure 7.

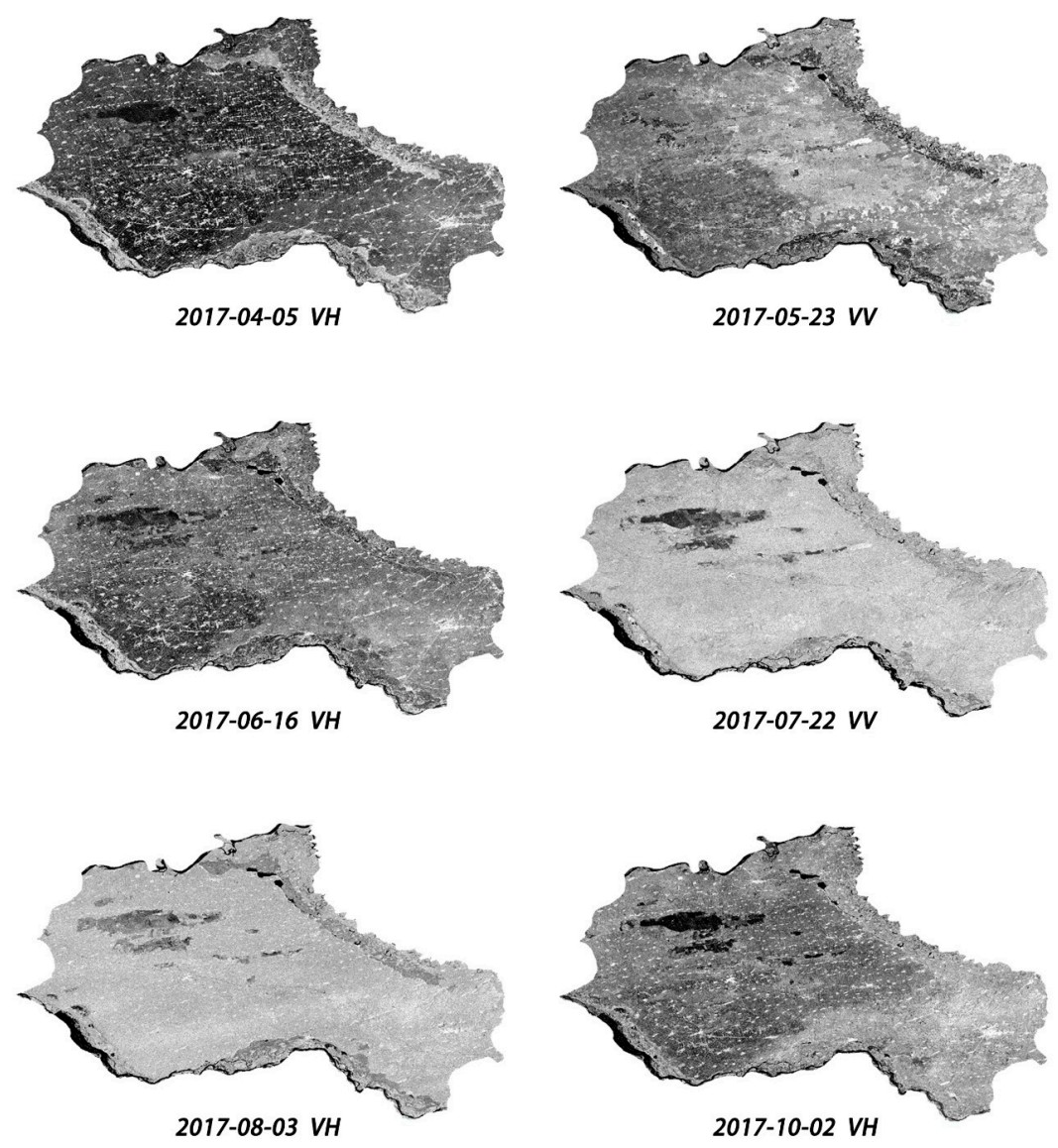

**Figure 7.** The optimal multi-temporal images.

*4.2. U-Net Model Training Details*

4.2.1. Training Samples

Based on the result of field survey combined with the Landsat-8 OLI data and Google Earth optical images, four 1000 × 1000 sub-regions were intercepted on the above processed data. The visual interpretation results are shown in Figure 8, and the distribution of per class of the training samples is shown on Table 5.

Since the SAR data is difficult and time-consuming to interpret visually by pixels, the number of training samples in the semantic segmentation network of SAR data is generally small, and the network training is prone to overfitting, which makes the test precision lower. Therefore, it is especially important to make changes to the sample. In this paper, the training samples and their labels were enhanced simultaneously by cutting, rotating, and flipping. Under the premise of not changing the SAR data backscatter coefficient, the number of training samples is multiplied by changing the spatial coordinates of the pixels and labels to improve the sample diversity. Through sample augmentation, the above four 1000 × 1000 sample sections were processed into 7840 × 224 × 224 sample sections.

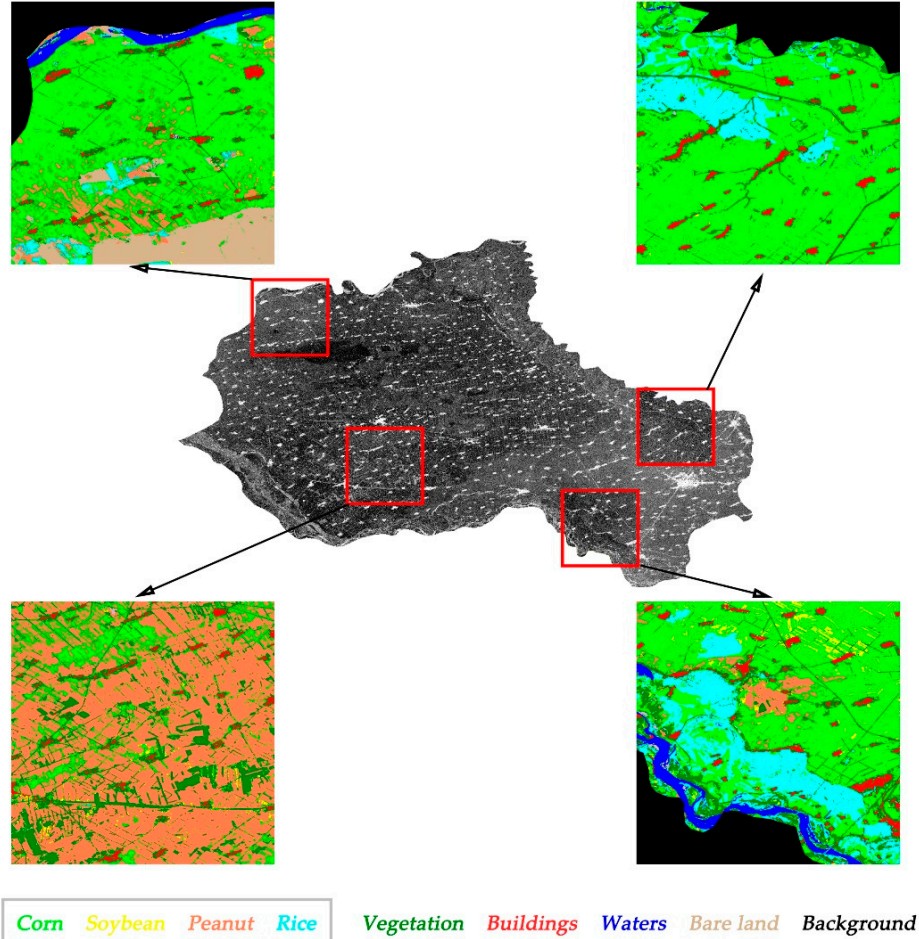

Corn  Soybean  Peanut  Rice     Vegetation  Buildings  Waters  Bare land  Background

**Figure 8.** Samples and labels for network training.

**Table 5.** The distribution of per class of the training samples.

| Value | Class | Pixel Count | Percent |
|---|---|---|---|
| 0 | Background | 194,626 | 4.87 |
| 1 | Buildings | 123,579 | 3.09 |
| 2 | Vegetation | 401,397 | 10.03 |
| 3 | Waters | 64,622 | 1.62 |
| 4 | Soybeans | 96,656 | 2.42 |
| 5 | Rice | 356,618 | 8.92 |
| 6 | Corn | 1,809,786 | 45.24 |
| 7 | Peanut | 806,050 | 20.15 |
| 8 | Bare land | 146,666 | 3.67 |
| total | | 4,000,000 | 100.00 |

#### 4.2.2. Training Details

In this paper, we use the network as shown in Figure 2. The input data is a 6-channel 224 × 224 SAR image slice with a labeling image. The training environment and parameters of the U-Net is shown on Table 6.

### 4.3. The Results of Crop Mapping

In Section 4.1, the multi-temporal images optimization method based on ANOVA and J–M distance was used to process the multi-temporal SAR data after preprocessing. As a result, the

dimension of network input is considerably reduced, which is significant when the hardware configuration is limited. In the training environment shown in Table 6, it is impossible to train the network with a 36-channel input. In Section 4.2, we used the optimal 6-channel multi-temporal image as network input to train the U-Net and predicted the crop mapping result of Fuyu City in 2017. In addition, we used RF and SVM classifiers to classify crops with the 36-channel multi-temporal image which the sample separabilities greater than 1.9 as a comparative experiment to the proposed method in this paper.

**Table 6.** The main parameters of U-Net network training.

| Training Environment | |
| --- | --- |
| CPU [1] | Core i7 |
| GPU [2] | NVIDIA GTX 1080Ti |
| Platform | Tensorflow |
| **Training Parameters** | |
| Input size | $224 \times 224 \times 6$ |
| batch-size | 5 |
| Learning rate | 0.001 |
| Total number of sample | 7860 |
| Epoch | 10 |

[1] CPU: Central Processing Unit. [2] GPU: Graphics Processing Unit.

Figure 9 shows the results of crop mapping in Fuyu City in 2017. It can be seen from Figure 9 that the results obtained by this method are roughly consistent with the field survey result introduced in Section 3.2.2. In addition, Figure 10 shows the distribution of the crop type statistics in 2014~2016, obtained from Statistic Bureau of Jilin Province (http://tjj.jl.gov.cn/) and the U-Net predicted result in 2017.

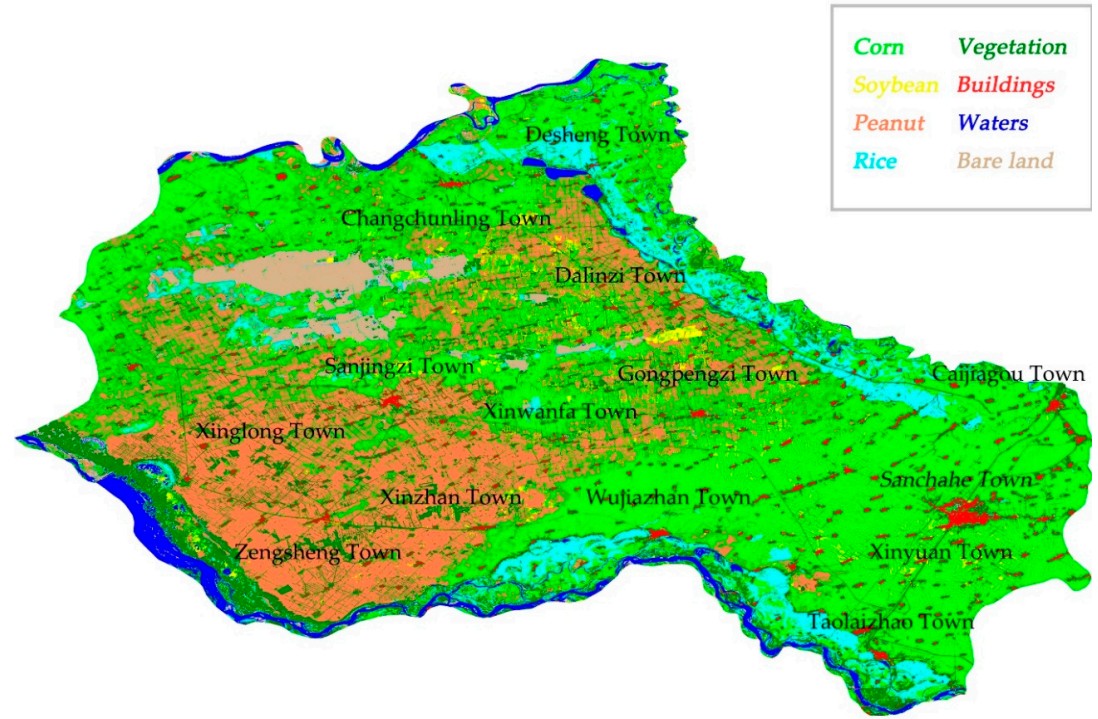

**Figure 9.** Crop mapping results of Fuyu City using the U-Net network.

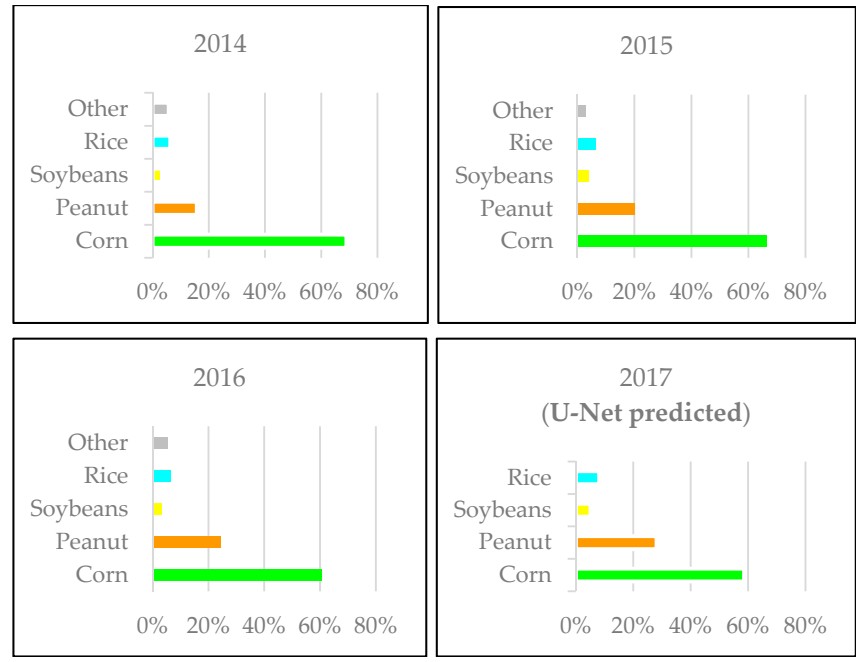

**Figure 10.** The distribution of the crop types of statistics in 2014–2016 and U-Net predicted result in 2017. The statistics were obtained from Statistic Bureau of Jilin Province (http://tjj.jl.gov.cn/).

In Figure 11, comparing the details of the crop mapping results in areas with complex planting structures, it is obvious that RF and SVM have more broken parts, but the U-Net can still maintain good classification results in these areas.

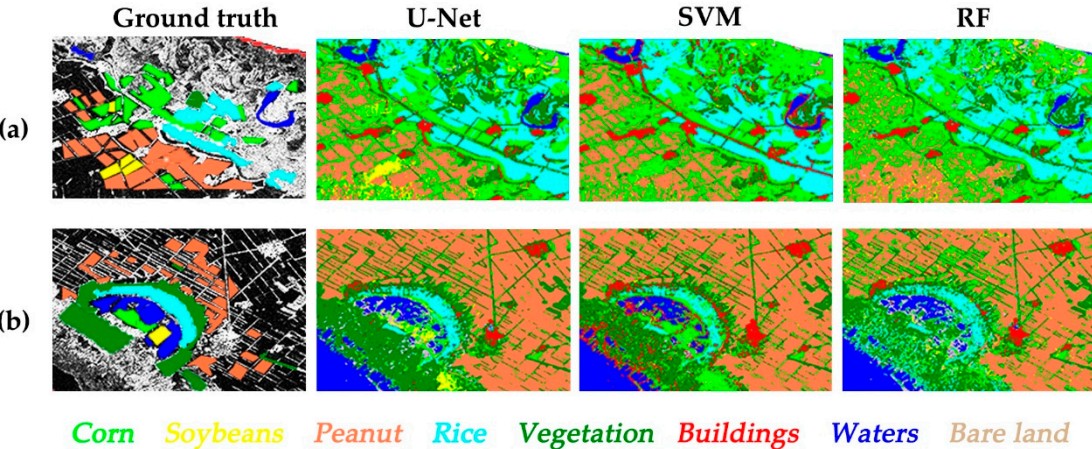

**Figure 11.** Crop mapping results of U-Net, SVM, and RF in areas with complex planting structures.

In addition to qualitative analysis, this paper uses the confusion matrix and Kappa statistics [45] to quantitatively analyze the prediction results of the U-Net model. Table 7 gives the overall accuracy (OA) and Kappa coefficients for the three classification methods. The overall accuracy of the three classification methods is above 75%, and the Kappa coefficient is greater than 0.70, which proves the feasibility of multi-temporal SAR crop classification. Compared with the traditional method, the U-Net model shows a significant improvement in the overall classification accuracy and Kappa coefficient. Furthermore, the commission error and the omission error of crop classification based on the U-Net model are smaller than those based on the traditional machine learning method, which shows that the U-Net model can still obtain the classification results better than the traditional machine learning method under the premise of using less data.

**Table 7.** Accuracy assessment of crop classification results based on confusion matrix.

| Class | RF | | SVM | | U-Net | |
|---|---|---|---|---|---|---|
| | Commission (%) | Omission (%) | Commission (%) | Omission (%) | Commission (%) | Omission (%) |
| Corn | 40.67 | 11.28 | 45.96 | 7.66 | 32.32 | 5.82 |
| Soybeans | 33.22 | 30.76 | 26.20 | 19.70 | 28.38 | 12.78 |
| Peanut | 20.52 | 21.34 | 12.98 | 13.97 | 12.43 | 14.64 |
| Rice | 12.59 | 30.43 | 0.71 | 37.23 | 1.48 | 24.52 |
| OA | 77.8582% | | 78.6219% | | 85.0616% | |
| Kappa | 0.7381 | | 0.7487 | | 0.8234 | |

Furthermore, the confusion matrix of the crop mapping result using U-Net was reported on Table 8. According to the confusion matrix, the findings were as follows: (1) the high commission error of corn ws caused by the misclassification of vegetation, soybeans, rice, and peanut to corn; (2) the high commission error of soybeans was caused by the misclassification of peanut and bare land to soybeans; (3) the high commission error of peanut was caused by the misclassification of rice and corn to peanut; and (4), the high omission error of rice was caused by the misclassification of rice to corn, peanut, and vegetation.

**Table 8.** Confusion matrix of the crop mapping result using U-Net.

| | Class | Build | Vegetation | Soybeans | Rice | Water | Peanut | Corn | Bare Land | Total |
|---|---|---|---|---|---|---|---|---|---|---|
| | | | | | **Ground Truth** | | | | | |
| | Build | 7805 | 424 | 3 | 84 | 0 | 0 | 10 | 0 | 8326 |
| | Vegetation | 8 | 37,683 | 147 | **2257** | 0 | 108 | 166 | 211 | 40,414 |
| | Soybeans | 0 | 791 | 11,744 | 246 | 0 | **1075** | 607 | **1927** | 16,372 |
| Predicted | Rice | 1 | 296 | 78 | 29,451 | 1 | 1 | 55 | 9 | 29,795 |
| | Water | 0 | 6 | 58 | 22 | 10,856 | 0 | 0 | 121 | 11,062 |
| | Peanut | 0 | 334 | 34 | **2163** | 0 | 33,514 | **1297** | 919 | 38,190 |
| | Corn | 4 | **4170** | **1208** | **4489** | 0 | **4423** | 35,462 | **2622** | 52,336 |
| | Bare land | 0 | 1773 | 172 | 160 | 15 | 60 | 10 | 22,311 | 24,422 |
| | Total | 7818 | 45,477 | 13,426 | 38,890 | 10,872 | 39,181 | 37,607 | 28,120 | 221,409 |

## 5. Discussion

The Sentinel-1 satellite can provide free SAR data in C-band, dual-polarization, multi-temporal and wide coverage, providing sufficient data for large-scale crop mapping. On the other hand, the successful practice of deep learning technology in SAR image processing provides technical support for large-scale and high-precision crop classification using SAR data. Therefore, it has important research significance and application potential to develop large-scale and high-precision crop classification research based on SAR data using deep learning methods. In this work, we introduced the deep learning semantic segmentation network, U-Net, to the multi-temporal Sentinel-1 SAR data crop mapping, and we obtained a satisfactory result with an overall accuracy of 85% and a Kappa coefficient of 0.82.

At present, this work obtained a favorable result, but it is still possible to improve in future work. (1) In this work, the temporal correlation of the multi-temporal data is not fully utilized, which account for the slightly disappointed commission error and omission error of crops. (2) The current classification system is simplified so that the predicted crop acreage slightly deviates from government statistics. Consequently, future work could be focused on making full use of the multi-temporal data to improve the accuracy of crop classification and establishing a more detailed classification system.

## 6. Conclusions

Dedicated to large area SAR image crop classification and mapping, this paper proposes a multi-temporal SAR crop mapping method based on an identification classification U-Net model. The main conclusions are as follows:

(a)　A multi-temporal images optimization method combining ANOVA and J–M distance is proposed to realize the reduction dimension of time series images, and to effectively reduce the redundancy of multi-temporal SAR data.

(b)　Through geometric transformation methods, such as cutting, flipping, and rotation, the diversity of samples is enhanced, and the network overfitting problem caused by fewer training samples is effectively solved.

(c)　The experimental results show that the multi-temporal SAR data crop mapping method based on the U-Net model can achieve higher classification accuracy under the condition of a complex crop planting structure.

**Author Contributions:** S.W. was mainly responsible for the construction of crop mapping dataset, conceived the manuscript, and conducted the experiments. H.Z. and C.W. supervised the experiments and helped discuss the proposed method, and also contributed to the organization of the paper; they also revised the paper and the experimental analysis. Y.W. and L.X. participated in the construction of the dataset.

**Funding:** This research was funded by the National Natural Science Foundation of China under Grants 41331176 and 41371352.

**Conflicts of Interest:** The authors declare no conflict of interest.

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
