# Peer review of "Multi-Temporal SAR Data Large-Scale Crop Mapping Based on U-Net Model"

_remotesensing, doi:10.3390/rs11010068_

Round 1

Reviewer 1 Report

It is necessary that the authors add or mention the few works where deep learning semantic segmentation technology has been used for crop classification, although the authors mentioned that this field has been rarely explored.

The manuscript Is original, however, what happen when the f-test values of the images are low?. The authors have to show an analysis taking into account the 6 images with the lowest F-test values or a mix of images with different F-test values, per example images with high F-test values and low F-test values. The aforementioned is in order to determine what happen if a study need to distinguish between several crops and the obtained image have not high F-test values. It can be used for training U-net model?

In the figure 9, the obtained classification is not clear, the regions are very small compared with the image, may be to make a zoom to those regions could help to observe the classified region and make a better qualitative comparison with the survey results.

The figure 10 can be improved by augmenting the zoomed regions of the images in order to compare better the crop mapping results using U-Net, SVM and RF.

The figure 11 can be improved in order to be more illustrative, the authors do not separate the regions obtained by using the surveys and google earth images and those obtained by using their methodology proposed in the manuscript, they could be more illustrative and show both, the regions obtained by the surveys and google earth and those regions obtained by using the methodology here proposed by the authors.

The authors need to justify more the advantages of having or reducing the dimension of time series images.

The authors mentioned that in the line 176 and 177 that 18 scenes from march to October were selected, however the authors mention a date of 20170204, it is supposed that this date is February, the authors need to correct this in the text and mention that the 18 scenes goes from February to October.

Author Response

We appreciate the thorough reviews provided by the referees and handling editor. We agree with these suggestions and have revised the manuscript accordingly. Below is our response to their comments resulting in a number of clarifications. We hope these revisions resolve the problems and uncertainties pointed out by the referees. In the manuscript and this file, the red and blue parts are revisions suggested by two referees, respectively. And, the magenta is revisions suggested by academic editor. The underline parts in the manuscript are those  changed contents to improve the expressions. 

Reviewer 2 Report

The manuscript submitted by S. Wei et al. “Multi-temporal SAR data Large-scale Crop mapping based on U-Net model ” proposes a methodology for the mapping multiple crop types over large area of China through the exploitation of a multi-temporal set of Sentinel-1 dual-polarization SAR data. Based on a deep learning approach originally developed in the Computer Vision domain, the proposed U-Net, an improved fully convolutional network, has been extended and adapted for the application to a Remote Sensing image classification problem.

This contamination from different research fields is interesting, however the manuscript could be considered for publication only after major revision.

Main concern refers to the accuracy assessment of thematic map output from the proposed approach.

The validation process to assess the accuracy of maps resulting from a classification should be an integral part of any of thematic mapping investigation. In this context, a statistically rigorous accuracy assessment requires choosing an appropriate sample size and sampling scheme, and considering the effects of spatial autocorrelation (Congalton, 1991, Remote Sensing of Environment 37, 35–4).

In order to assure a statistically rigorous evaluation of entire population, it is essential that randomization is incorporated into the selection of test samples (sample selection protocol).

Although practical constraints of an experiment should be considered in the design, “All deviations from the probability sampling protocol should be documented and quantified to the greatest extent possible.”(Olofsson et al., 2014, Remote Sensing of Environment 148, 42–57).

All these elements should be considered, while in the paper it is only limited to Fig. 11 and the sentence in Line 293.

Finally together with the  summary  metrics  of  accuracy (Table 4), the  raw confusion  matrix  should  be  presented at least for the U-Net.

ANOVA is a set of statistical models and associated estimation procedures that are well-known to researchers working in the classification of remotely sensed data. In the paragraph 2. Methods, one full page is dedicated to Analysis of Variance (2.1.1) with seven equations, while some specific innovations of this paper such as  “multiplication” of training and Batch Normalization (BN) algorithm are presented in the Results and should be moved to Methods.

Other comments:

Line 33: Suggest to reverse this sentence: largest population vs small arable land.

Line 55-56: Common SAR data should include the Cosmo SkyMed constellation (suggested ref: Villa et al. Remote Sensing 2015, 7, 12859-12886; doi:10.3390/rs71012859)

Line 72: “Sergey et al. “ not present in References; [26] is Ronneberger et al.

3. Study area and data

This paragraph, which describes about arable land extension, does not contain any information about “smallholder farming modes, which caused the complex crop-planting patterns” mentioned in the Abstract

Figure 3: please convert “Miles” to “Km”

Line 176: “... 18 scenes from March”, first image in Table 1 is February as shown also in Fig. 1.

Line 183-184 “According to the survey results of land cover types”; give more information on ground surveys / field investigations

Line 187-189 and Fig. 4: What does mean “appropriate samples” ? Temporal profiles refer to a specific pixel or a set of pixel ? How many pixels for each 8 land cover type, from the same site or from different? Are they uniform (mean, St. Dev).

Line 196: “It confirms the feasibility ..”; This cannot be derived from the graph (e.g. Corn and soybean profiles show similar trend), but only from ANOVA and J-M distance.

Line 218-223 and Fig. 5: If 1.8 is “threshold” J-M distance value can be useful to put in the graph a dashed line in correspondence to this specific value. Explain the reason why you selected 6 images and not 5 images or 7.

Line 251: “the sample set is uneven”; the authors should be more specific indicating the number of pixel selected for each of the 8 land cover categories.

In this paragraph 4.3 the terms “feature” “category” are used in a mixed way and this does not  contribute to clarity. Usually category, class, land cover type are used with same meaning, while feature indicates characteristics or attribute/quality associated to a category.

4.4. An analysis of the classification results

Beside the thematic map of Fig 9, first results interesting to know are quantitative extension occupied by each of the 8 land cover (both value and %): this information will help to understand the importance of each class in particular for crops.

Line 264 & following “Through the investigation of crop planting …” Which is the basis for this investigation ? Should this type of information be anticipated in the description of Study area ? 

Line 304-305. “The reason for the high commission error is the difference in the crop planting time as well as the effect of other vegetation and other ground objects other than the above four crops.”

It is suggested to be less generic, e.g. difference in crop planting for which crop type. Moreover presentation of raw confusion matrix could help in indicating which class (crop) is confused with which other.

Other terminology mix-up:

“crop planting” and “crop mapping” are synonyms ?

“ground object” vs “land cover”

Author Response

(The authors gave the same response as above.)

Round 2

Reviewer 2 Report

The Authors replayed almost satisfactory to all my comments, some minor revisions are suggested.

 The use of Plot as reference data in the validation process requires a comment / consideration on the spatial autocorrelation, such as for example a sentence similar: “Surely these results should affect the sample size and especially the sampling scheme used in accuracy assessment, especially in the way this autocorrelation affects the assumption of sample independence “ (Congalton, 1991, Rem. Sens. Environ., n. 37, page 43).

Line 180 “Sergey Ioffe and Christian Szegedy in 2015 [43].”  Change to standard form “Ioffe and Szegedy in …”

Line 220 “ Landsat-8 OLI data in 2017”    Could you specify how many Landsat images and which dates ?

Fig. 10: First image of 2014, x-axis reports values as decimal point, while the other three images (2015, 2016, 2017) are expressed in %. Please uniform

Line 349 “In Figure 10, comparing the details of the crop mapping results” This should be Figure 11

Author Response

We appreciate the thorough reviews provided by the second referee. We agree with these suggestions and have revised the manuscript accordingly. Below is our response to these comments resulting in a number of clarifications. We hope these revisions resolve the problems and uncertainties pointed out by the referee. In the manuscript and this file, the red parts are revisions suggested by the referee, respectively. The underline parts in the manuscript are those the changed contents to improve the expressions. 
